# Analysis of the Postharvest Storage Characteristics of Two New Pear Cultivars 'Shannongsu' and 'Xincixiang'

Susu Zhang [1,†], Sumin Qi [1,2,†], Bin Li [3], Jing Zhang [1], Can Cui [1], Rui Zhang [4], Zhiquan Mao [1], Nan Wang [1], Xuesen Chen [1,*] and Zongying Zhang [1,*]

1   State Key Laboratory of Crop Biology, College of Horticulture Sciences, Shandong Agricultural University, No. 61 Daizong Road, Tai'an 271018, China
2   Shandong Institute of Pomology, No. 64 Longtan Road, Tai'an 271000, China
3   Lin'yi Forestry Bureau, 11 Beijing Road, Lin'yi 276000, China
4   College of Agriculture and Bioengineering, Heze University, No. 2269, University Road, Mudan District, He'ze 274015, China
*   Correspondence: chenxs@sdau.edu.cn (X.C.); zongyingzhang@sdau.edu.cn (Z.Z.)
†   These authors contributed equally to this work.

**Abstract:** 'Shannongsu' and 'Xincixiang' were two new late-ripening pear cultivars with high quality, that were bred by our team. In order to clear the postharvest storage characteristics, mature pears were collected and stored at room temperature. Several fruit characteristics were evaluated over time, such as firmness, ethylene release rate, content of aroma components, softening-related enzyme activity, and gene expression. Both 'Shannongsu' and 'Xincixiang' were crisp and juicy stored after 60 d at room temperature, which exhibited excellent storage characteristics. The storability may be comprehensive result of low levels of ethylene production, aroma synthesis, softening-related activities, and gene expression. The research cleared the storage characteristics of 'Shannongsu' and 'Xincixiang' at room temperature for the first time, which will provide scientific theoretical guidance for fruit production and marketing.

**Keywords:** pear; breeding; ethylene; fruit softening; storage quality





## 1. Introduction

Pear is one of the most important fruits in temperate latitudes, which is widely cultivated around the world [1]. China is the largest producer, consumer, and exporter of pears, and a number of traditional cultivars are well known at home and abroad, such as 'Korla', 'Dangshansu', 'Yali', and 'Laiyangci' pear. In recent decades, many early-maturing cultivars were bred, such as 'Huangguan', 'Cuiguan' [2,3]. However, as seasonal fruits, early-maturing cultivars cannot act as the main cultivars. Therefore, traditional cultivars, such as 'Yali' and 'Dangshansu', remain the main cultivars in China, which make up more than 60% of the planting area [4,5]. With the improvement of consumption level, traditional cultivars cannot meet the elevated demand of consumers. The progress of the pear industry is facing pressures and challenges, and it is urgent to transform and upgrade. As a large consumption fruit in China, it is economic to plant and sell high-quality late-maturing pear cultivars [6]. However, few such new varieties are available currently. Therefore, it is critical to breed high-quality late-maturing pear cultivars to meet the large consumer market in China.

It is well known that 'Dangshansu', 'Laiyangci', and 'Korla' pears are three traditional, famous cultivars in China, whereas the quality needs to be improved as the improvement of living standards. Therefore, cross-breeding experiments were carried out by our team in 2003. In terms of a patented technology-multiple species quality breeding method of fruit trees, 'Dangshansu', 'Laiyangci', and 'Xinli No.7', were selected as crossing parents [4]. Twelve years later, two new cultivars named 'Shannongsu' and 'Xincixiang' were selected

and approved from different crossing combinations. Both of the two cultivars have excellent fresh quality, especially 'Shannongsu', which is late-ripening and not prone to browning, with a small core, crispness, high content of soluble solids (12.7%), few stone cells, long shelf-life [7,8]. All the quality characteristics meet the requirement of consumers in China, and it is helpful to realize the annual supply in high-level market. Therefore, 'Shannongsu' has become the first choice in the planting of pears, and the area has been over 3300 hectares in the past five years. The sale price is over RMB 40 yuan per kilometer, and the excellent quality is widely welcomed by consumers.

Storage quality is the key to transform from fruits to commodities. Therefore, it has been the focus of many studies. Ethylene is well known as an important plant hormone involved in fruit ripening, softening, and senescence. The shelf-life of fruits is regulated by ethylene, especially climacteric fruits [9]. ACS (1-aminocyclopropane-1-carboxylic acid synthase) and ACO (1-aminocyclopropane-1-carboxylic acid oxidase) are the key enzymes involved in ethylene biosynthesis. Inhibiting the expression of the ethylene biosynthesis genes for ACS and ACO will inhibit ethylene biosynthesis in fruit, and also slow down the process of fruit ripening and softening, thus effectively prolonging the shelf-life [10–12]. Fruit softening is generally considered the result of destruction and degradation of the cell wall, which results from the action of various cell wall-degrading enzymes, such as polygalacturonase (PG), xyloglucan endotransglycosylase (XET), $\beta$-galactosidase ($\beta$-Gal), $\alpha$-L-arabinofuranosidase ($\alpha$-L-Af), and pectin methylesterase (PME) [13,14]. Though there are many studies about cell wall-degrading enzymes, the role of key enzymes is different in species and cultivars. As new cultivars with long shelf-life, 'Shannongsu' and 'Xincixiang' may be ideal experimental materials for studying fruit ripening and softening.

Therefore, the postharvest storage characteristics at room temperature of the two cultivars were analyzed, including the production of ethylene and aroma volatile, the activities and expressions of the enzymes related with fruit softening, which aimed to better understand the mechanism of fruit softening and provide scientific bases for fruit breeding and production.

## 2. Materials and Methods

### 2.1. Materials

The experiment was carried out in the Key Laboratory of Crops of Shandong Agricultural University. 'Shannongsu' and 'Xincixiang' pears were collected from the fruit breeding base in Liaocheng on 15 October 2017, while 'Whangkeumbae' pears were collected from the Yantai on 16 September 2017. Mature pear fruits without plant diseases, insect pests, and mechanical scars were collected and immediately transferred to the laboratory for storage experiment at room temperature (25 °C). Five pears were selected to determine relevant indicators of ethylene release rate every 3 days. Then, pear flesh samples were frozen in liquid nitrogen and stored in −80 °C freezer.

### 2.2. Methods

#### 2.2.1. Determination of Fruit Firmness

Fruit firmness was measured by a texture analyzer (Stable Microsystems, Godalming, UK) with the whole-fruit puncture method [14]. The conditions of the texture analyzer were set as follows: P/2 columnar probe, the pre-measurement, measurement, and post-measurement speeds were 2 mm·s$^{-1}$, 1 mm·s$^{-1}$, and 5 mm·s$^{-1}$, respectively, the puncture depth was 10 mm, and the minimum sensing force was 10 g. Four fruits of each cultivar were randomly selected for the measurement, and each fruit was measured four times around the equatorial plane.

#### 2.2.2. Determination of Ethylene Release Rate

The ethylene release rate was measured by Shimadzu GC-9A gas chromatograph (Kyoto, Japan). Pears were sealed in a glass container and stored at room temperature for 6 h before the measurement. Next, 1 mL of gas was extracted from the glass container and

injected into the gas chromatograph inlet. The gases were analyzed using the following settings: Detector and separation column temperatures of 140°C and 70 °C, respectively, $N_2$, $H_2$, air, and make-up gas flow rates of 25, 40, 400, and 15 mL·min$^{-1}$, respectively. The ethylene release rate was calculated by quantifying the peak area from the GC [14].

### 2.2.3. Extraction and Detection of Fruit Aroma Components

The fruit aroma compounds were extracted and detected as described by Wang [15]. The aroma compounds were detected using a GC/MS-QP2010 gas chromatography-mass spectrometer produced by Shimadzu. The detection sequence was as follows: The fiber extraction head was inserted into the GC inlet at 250 °C then aged for 30-50 min. An SPME (solid phase microextraction) fiber coated with DVB/CAR/PDMS (divinylbenzene/carboxen/polydimethylsiloxane) was used to extract at 50/30 μm. The samples were washed and dried, then 20 g samples of the pears were weighed and crushed using a blender. As the internal standard, 5 μL 3-nononone (0.1 mg·mL$^{-1}$) with the pulp and juice were added into a conical flask. The conical flask was placed on a heating plate at 45 °C with magnetic stirring for 40 min, and the aged extraction head was inserted into the conical flask for the extraction. Then, the extraction head was inserted into the GC inlet for 2 min and then analyzed by GC-MS. Qualitative and quantitative methods were used to identify and quantify the aroma substances.

### 2.2.4. Determination of Enzyme Activities

The method of preparing the enzyme solution was improved. As described by Zhou [16]. Pear flesh (5 g) was ground into powder in liquid nitrogen. The extraction solution was a 0.1 M sodium acetate buffer (pH 5.2) containing 0.1 M NaCl and 1.5% polyvinyl pyrrolidone (PVP). The activity of XET was determined as described by Percy [17]. Using 3 mM p-nitrophenyl-α-L-arabinofuranoside and p-nitrophenyl-β-D-galactopyranoside as substrates, the activities of β-Gal and α-L-Af were determined as described by Brummell [18]. Referring to Gross's method, the activity of polygalacturonase (PG) was determined using 0.5% polygalacturonic acid as substrate, and the absorbance value was measured at 276 nm by a photometer. The activity of PME was determined as described by Liu [19].

### 2.2.5. RNA Extraction and qRT-PCR Analysis

The total RNA was extracted from the pulp using a DP441 RNAprep Pure Plant Plus kit (Tiangen Biotech Co., Ltd., Beijing, China). The cDNA was synthesized by reverse transcription using RNA as the template. The cDNA was synthesized using the Trans Script One-Step gDNA Removal and cDNA Synthesis SuperMix kits (Trans Gen Biotech, Beijing, China). The qRT-PCR used full-mode Gold Trans Start Top Green qPCR Super Mix kit (Trans Gen Biotech), with the pear actin gene as an internal reference. The PCR program was as follows: 94 °C for 30 s; 40 cycles of 94 °C for 5 s, 55 60 °C for 15 s, and 72 °C for 10 s; and 72 °C for 5 min. The relative gene expression levels were calculated by the $2^{-\Delta\Delta Ct}$ method [20].

## 3. Results

### 3.1. Changes in Fruit Firmness

Fruit firmness of 'Shannongsu' 'Xincixiang', and 'Whangkeumbae' all decreased during storage time at room temperature (Figure 1). Fruit firmness of 'Whangkeumbae' quickly decreased during storage time, and it decreased by 71.8% after 15 d of storage, showing a cotton-like texture (it lost measurement value after 15 days of storage). By contrast, fruit firmness of 'Shannongsu' and 'Xincixiang' only decreased by 26.6% and 23.4%, respectively, after 60 d of storage, and fruits were still crisp and juicy (0.12 kg/cm$^2$).

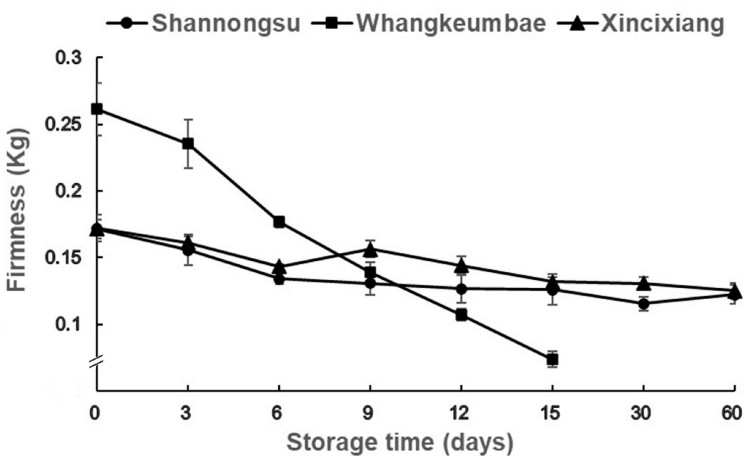

**Figure 1.** Changes in fruit firmness of three pear cultivars stored at room temperature. Values shown represent means ± SD (n = 3).

### 3.2. The Production of Ethylene

As shown in Figure 2, the peak of the ethylene release rate emerged on the harvest day among the three cultivars. The ethylene release rate of 'Whangkeumbae' was 3.64 µL/h/g FW on the day of harvest, which was 3–7 times higher than that of 'Shannongsu' and 'Xincixiang'. During storage time, the ethylene release rate of 'Shannongsu' and 'Xincixiang' was always at a very low level, and it was much lower than that of 'Whangkeumbae' all the time. There was a significant negative correlation between ethylene release rate and fruit firmness in 'Whangkeumbae' ($-0.92$, $p < 0.01$), while there was no significant correlation ($p > 0.05$) in 'Shannongsu' and 'Xincixiang'. As the key genes of ethylene biosynthesis, the variation trend of ACS and ACO expression was similar to the ethylene release rate, and the expression of 'Shannongsu' and 'Xincixiang' was significantly lower than that of 'Whangkeumbae' all the time.

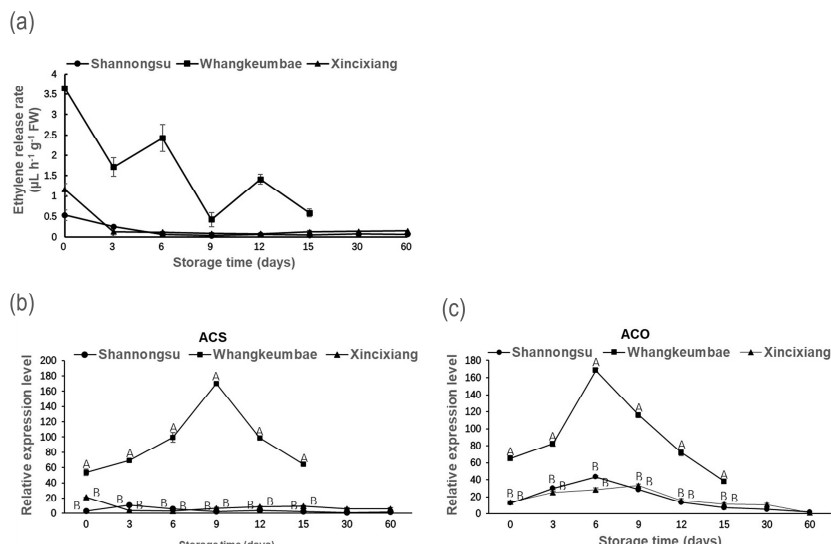

**Figure 2.** (**a**) Ethylene release rate in three pear cultivars; (**b**) Gene expression of ACS relative expression levels in three pear cultivars; (**c**) Gene expression of ACO relative expression levels in three pear cultivars. Values shown represent means ± SD (n = 3). Different letters indicate significant differences between treatments ($p \leq 0.01$).

### 3.3. Fruit Softening-Related Enzyme Activity and Gene Expression

The activity of enzymes related to fruit softening was much lower in 'Shannongsu' and 'Xincixiang' than that in 'Whangkeumbae' all the time, such as PG, PME, XET, α−L−Af,

and β−Gal (Figure 3). Most of them reached the peak of enzyme activity after 6 d of storage in 'Whangkeumbae', whereas fruit firmness quickly decreased during 0–6 d of storage. In contrast, the activity of enzymes did not show significant variation in 'Shannongsu' and 'Xincixiang' during storage time.

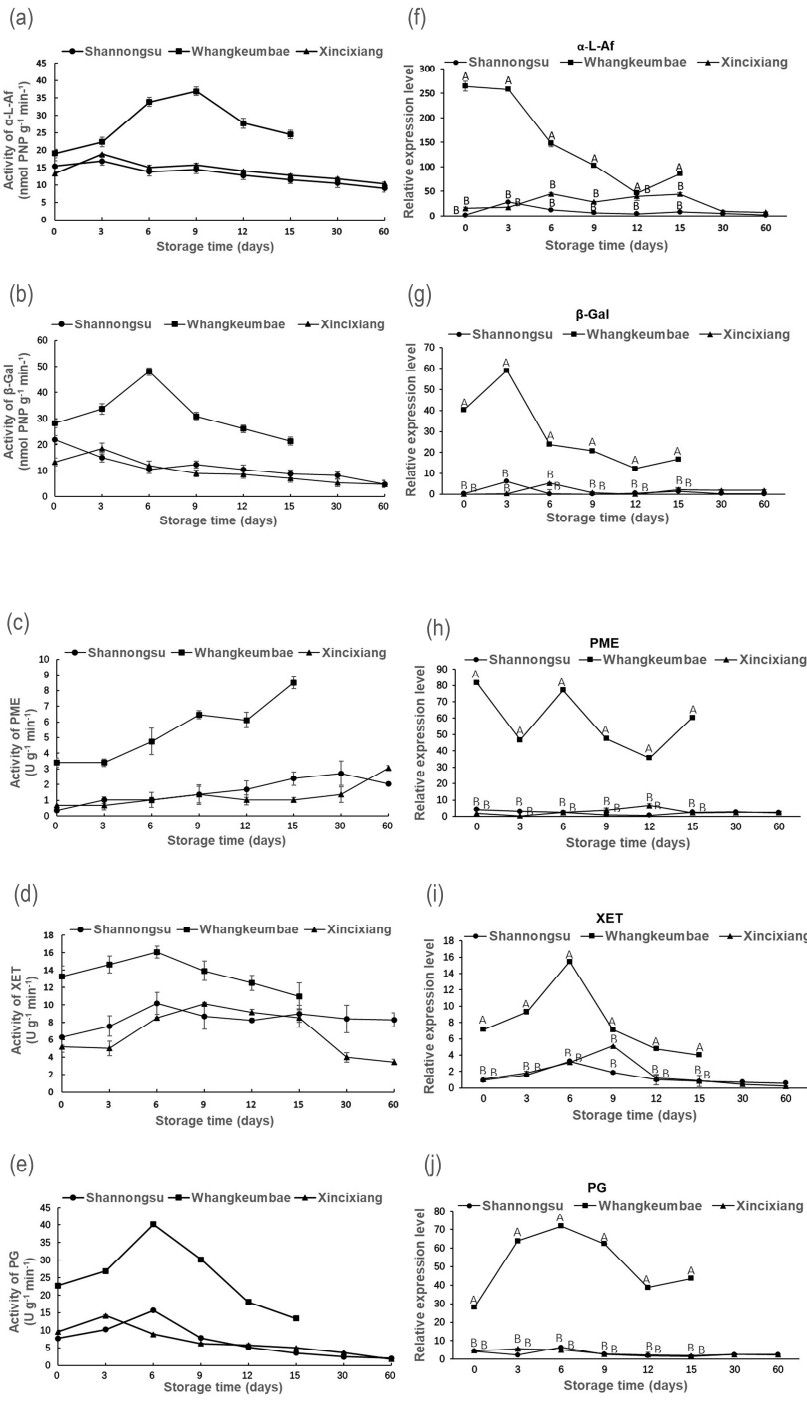

**Figure 3.** Fruit softening-related enzyme activity and gene expression in three pear cultivars. (**a**) α−L−Af activity, equivalence in p-nitrophenol; (**b**) β−Gal activity, equivalence in p-nitrophenol; (**c**) PME activity, equivalence in d-galacturonic acid; (**d**) XET activity; (**e**) PG activity, equivalence in d-galacturonic acid; and (**f**–**j**) Gene relative expression levels of α−L−Af, β−Gal, PME, XET and PG at room temperature. Values shown represent means ± SD (n = 3). Different letters indicate significant differences between treatments ($p \leq 0.01$).

The expression of key gene-encoding enzymes was significantly higher in 'Whangkeumbae' than that in 'Shannongsu' and 'Xincixiang', especially during early storage time (0−9 d). There was an obvious peak of expression in 'Whangkeumbae' during 0−9 d of storage, while there was no significant variation in 'Shannongsu' and 'Xincixiang' during 0−60 d of storage.

### 3.4. The Contents of Aroma Volatile and Expression of Key Gene Related to Aroma

The contents of primary aroma volatile were compared among the three pear cultivars, such as alcohol, aldehyde, and ester (Figure 4). The contents of aldehydes and alcohols in 'Shannongsu' and 'Xincixiang' were at a low level and significantly higher than those in 'Whangkeumbae' during the whole storage time. On the contrary, the contents of esters in 'Shannongsu' and 'Xincixiang' were kept at a low level during storage time and were significantly lower than those in 'Whangkeumbae' during storage time, which was only 8%, 33% of those in 'Whangkeumbae'.

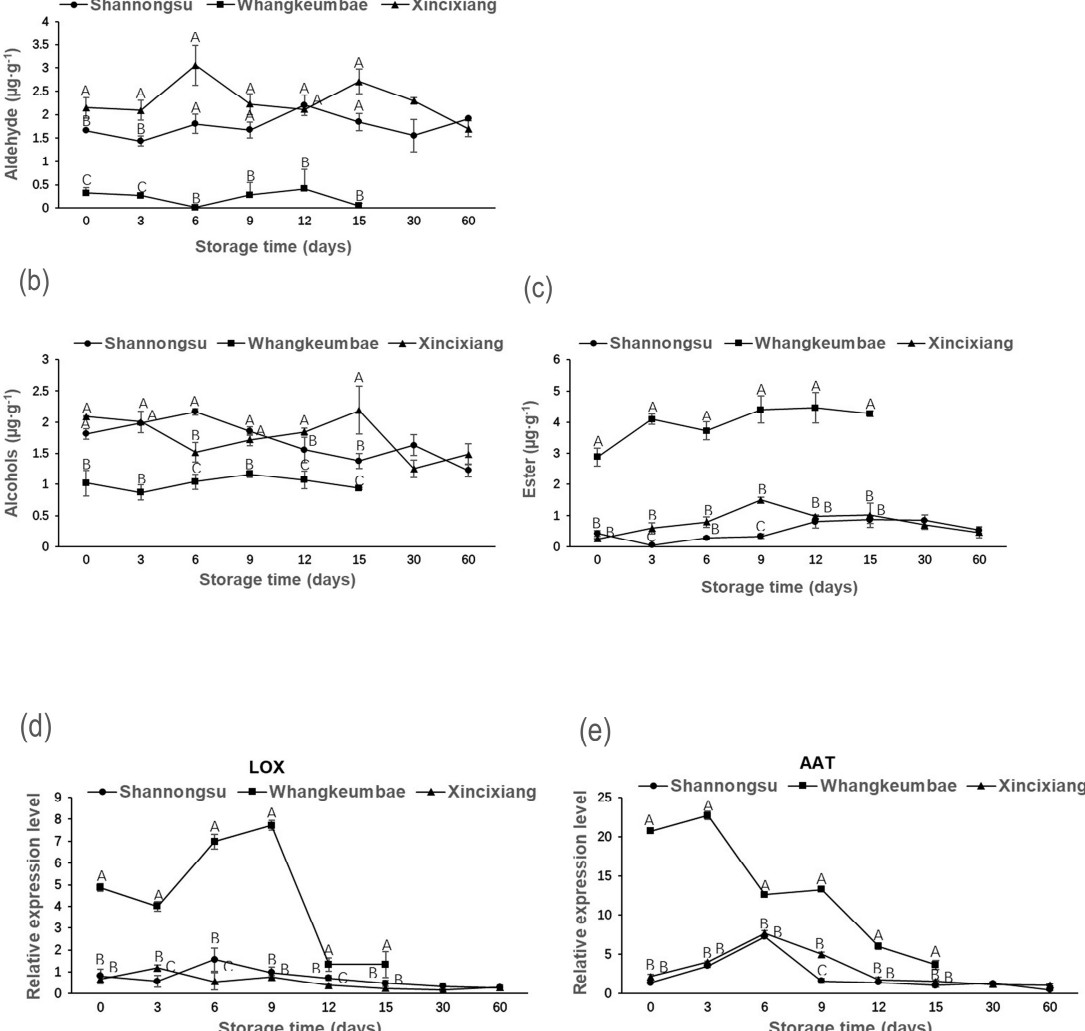

**Figure 4.** The contents of aroma volatile and expression of key genes related to aroma. (**a**–**c**) Components of aroma; (**d**,**e**) expression of key genes related to aroma production. Values shown represent means ± SD (n = 3). Different letters indicate significant differences between treatments ($p \leq 0.01$).

As the key genes related to aroma biosynthesis, the expression pattern of LOX and AAT was similar in three cultivars, with a trend of decreasing after increasing. The difference

was the expression of LOX and AAT in 'Whangkeumbae' was significantly higher than that in 'Shannongsu' and 'Xincixiang', which was 2–8 times, 2–15 times higher than that in 'Shannongsu' and 3–12 times, 2–10 times higher than that in 'Xincixiang', respectively.

## 4. Discussion

### 4.1. The Pivotal Role of Ethylene in Regulating Postharvest Storage Quality

Ethylene is well known as one of the most important plant hormones in regulating fruit ripening and softening [9]. Aminocyclopropane-1-carboxylate synthase (ACS) and aminocyclopropane-1-carboxylate oxidase (ACO) are the rate-limiting enzymes in the biosynthesis of ethylene. The two enzymes were encoded by ACS and ACO genes, and the expression of ACS and ACO was closely related to ethylene production and shelf-life [21]. In this study, the expression of ACS and ACO was kept at a lower level in 'Shannongsu' and 'Xincixiang' than that in 'Whangkeumbae' during 60 d of storage, resulting in less ethylene production in the two cultivars, which suggested ethylene might be the key regulator of excellent storage quality upstream in 'Shannongsu' and 'Xincixiang'. 1-methylcyclopropene (1-MCP) can inhibit ethylene action via binding to the ethylene receptor, which has been widely used for prolonging shelf-life in fruits and vegetables [22,23]. Conversely, ethephon is widely used for promoting fruiting ripening [24]. Therefore, the role of ethylene in regulating postharvest storage quality could be further verified by treatment of 1-MCP and ethephon.

Changes in the cell wall structure and composition are the direct cause of fruit ripening and softening, which are resulted from multiple hydrolytic enzymes of cell wall modification [20,25]. In this research, the rapid softening and elevation of enzyme activity and gene expression in 'Whangkeumbae' contrasted starkly with little change in 'Shannongsu' and 'Xincixiang', which suggested cell wall modifying enzymes (PG, PME, $\alpha$-L-AF, $\beta$-Gal, and XET) may be the direct cause of fruit softening. Moreover, the variation of enzyme activities and gene expression related to cell wall modification was closely associated with ethylene production, which suggested cell wall modifying enzymes may be regulated by ethylene. Therefore, the excellent storage quality of 'Shannongsu' and 'Xincixiang' may result from low enzyme activities which may be regulated by ethylene.

### 4.2. The Relationship between Aroma Volatile and Storage Quality

The composition and content of fruit aroma volatile are not only important factors affecting the consuming desire but also closely associated with fruit storage quality [26]. Esters are generally the primary aroma components in varieties with short-storage character, while aldehydes and alcohols are the main components of cultivars with long-storage character [19,27,28]. Treatment of 1-MCP can inhibit the production of ethylene and esters and maintain better storage quality [14,29,30]. In the study, we found a low concentration of esters and a high concentration of alcohols and aldehydes in 'Shannongsu' and 'Xincixiang' were lower all the time during storage time, which further verified the composition and content of aroma volatile were closely related to storage quality. The production of esters is in contradiction with the maintaining of fruit firmness in most fruits during storage time. Therefore, it is critical to balance the two factors well via breeding or technical approaches.

### 4.3. The Exploitation and Utilization of Excellent Germplasm Resources

China is the origin center and genetic diversity center of pear, with a variety of wild and local resources. The results of parental traceability analysis of high-quality fruits show the complex genetic background and rich genetic diversity are critical for cross-breeding [4,31]. 'Dangshansu' (P. bretschneideri Rehd), 'Laiyangci' (P. bretschneideri Rehd), and 'Korla' (P. sinkiangensis Yu) are three famous traditional cultivars of pear, which are crisp, juicy and storable. 'Xinli No.7', the female parent of 'Shannongsu' and 'Xincixiang', which is a high-quality cultivar selected from 'Korla'. The comprehensive quality characters of 'Shannongsu' and 'Xincixiang' pears are excellent. The average fruit weight of 'Shannongsu' is 460 g, with large fruit and small heart. The meat is fine, crisp, and of high quality. The fruit

does not brown after being cut for 24 h and is antioxidant [32]. 'Xincixiang' pear has white and delicate flesh, juicy, sweet taste, unique aroma, and excellent quality [8]. Therefore, 'Shannongsu' ('Xinli No.7' × 'Dangshansu') and 'Xincixiang' ('Xinli No.7' × 'Laiyangci') inherit the lineage of four species, such as P. sinkiangensis Yu, P. pyrifolia (Burm. f.) Nakai, P. communis L., P. bretschneideri Rehd., which is the key to outstanding quality [4]. The breeding of 'Shannongsu' and 'Xincixiang' further verifies the importance of multiple species quality breeding method, which suggests it is critical to trace the origin of cultivars. Therefore, it may be more efficient to select cultivars with complex genetic backgrounds and rich genetic diversity in cross-breeding.

## 5. Conclusions

'Shannongsu' and 'Xincixiang' pears can be stored for 60 d at room temperature without a noticeable reduction in quality. The low level of ethylene may be the key to the outstanding storage characteristics in 'Shannongsu' and 'Xincixiang'. The low level of fruit softening-related enzyme activity and gene expression (PG, PME, XET, $\alpha$-L-Af, and $\beta$-Gal) may be the direct cause of excellent storage quality. Moreover, the low level of esters regulated by LOX and AAT genes is also closely associated with excellent storage quality.

**Author Contributions:** Conceptualization, S.Q.; methodology, J.Z. and C.C.; B.L.; data curation, C.C.; writing—original draft preparation, S.Z.; writing—review and editing, Z.Z.; visualization, R.Z., Z.M. and N.W.; supervision, N.W.; project administration and funding acquisition, Z.Z.; funding acquisition, X.C. All authors have read and agreed to the published version of the manuscript.

**Funding:** This research was funded by the Special Fund Project of the Central Government in Guidance of Local Science and Technology Development, the Key R&D Plan of Shandong Province and the Guanxian Rural Revitalization Science and Technology Cooperation Fund, grant number YDZX2022103, 2022TZXD0037 and GXJJ2020-01.

**Data Availability Statement:** The authors do not have permission to share data.

**Conflicts of Interest:** The authors declare no competing financial interest.

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
