# Peer review of "Analysis of the Postharvest Storage Characteristics of Two New Pear Cultivars ‘Shannongsu’ and ‘Xincixiang’"

_horticulturae, doi:10.3390/horticulturae9020281_

Round 1

Reviewer 1 Report

The manuscript discusses post-harvest characteristics of two new pear cultivars. The charateristics and ripening mechanisms are based on knowledge already established in the literature and does not provide a signficant or novel contribution, other than confirmation of underlying mechanisms in the new pear varieties.

The manscuript lacks some information to better understand the data e.g. which variety was used as comparator and why was the analysis of the comparator stopped at 15 days, contrary to the 60 days analysis of the new varieties.

3.4. Describe how you established the values for ester, aldehyde and alcohols. Did you add the concentrations of several different compounds? How many individual compounds did you identify for each chemical class?

4.3 This paragraph does not seem based on scientific facts. How did the authors establish that the two varieties are outstanding? Is this solely based on the ripening? What is the consumers’ perception of these varieties?

Figure1. It is almost impossible to distinguish between the symbols in the graph.

Line 181: use (P<0.05) instead of at the 0.05 level.

The Enlgish needs to be corected to a more scietific standard and checked for spelling mistakes (e.g. Line 16: I presume it should be bred instead of bed).

Author Response

Response to Reviewer 1 Comments

Dear editor,

Thank you very much for your comments and suggestions. Those comments are all valuable and very helpful for revising and improving our paper, as well as the important guiding significance to our researches. We have studied comments carefully and have made correction which we hope meet with approval. The main corrections in the paper and the responds to the reviewer’s comments are as follows:

1.Comment: The manuscript discusses post-harvest characteristics of two new pear cultivars. The charateristics and ripening mechanisms are based on knowledge already established in the literature and does not provide a signficant or novel contribution, other than confirmation of underlying mechanisms in the new pear varieties.

Response: Thanks for your kind question. The postharvest storage characteristics at room temperature of the two cultivars were analyzed, including the production of ethylene and aroma volatile, the activities and expressions of the enzymes related with fruit softening, which aimed to better understand the mechanism of fruit softening and provide scientific bases for fruit breeding and production.

2.Comment: The manscuript lacks some information to better understand the data e.g. which variety was used as comparator and why was the analysis of the comparator stopped at 15 days, contrary to the 60 days analysis of the new varieties.

Response: Thanks for your kind question. As the control cultivar, ‘Whangkeumbae’ pear soften after 15 days of storage and lost the commodity value, while ‘Shannongsu’ ‘Xincixiang’ pears did not appear this phenomenon until 60 days. Therefore, the storage experiment of ‘Whangkeumbae’ stopped at 15 days. " It lost measurement value after 15 days of storage " is added in line 144 of the text to better describe the results.

3.Comment: 3.4. Describe how you established the values for ester, aldehyde and alcohols. Did you add the concentrations of several different compounds? How many individual compounds did you identify for each chemical class?

Response: Thanks for your kind question. In future, the specific ingredients of aroma may be studied in another article. However, in this article, our research focused on the total amount of aroma, not a specific substance, which was closely related with fruit storage quality. The fruit quality is reflected by the total content of ester aroma substances. This method has been used in many articles published by our laboratory, such as:

(1)  Lu L, et al. Analysis of the postharvest storage characteristics of the new red-fleshed apple cultivar 'meihong'. Food Chemistry, 2022, 354:129470.

(2) Liu JX, et al." Changes of firmness, aroma, cell wall-modifying enzyme activities and analysis of related-gene expression in 2 red flesh apple strains during fruit storage." Acta Horticulturae Sinica, 2017, 44(2): 330-342.

  1. Comment: 4.3 This paragraph does not seem based on scientific facts. How did the authors establish that the two varieties are outstanding? Is this solely based on the ripening? What is the consumers’ perception of these varieties?

Response: Thanks for your kind question. According to your suggestion, the following contents have been added in 4.3: ‘The average fruit weight of ‘Shannongsu’ is 460g, with large fruit and small heart; The meat is fine, crisp and of high quality; The fruit does not brown after being cut for 24 hours, and is antioxidant [40]. ‘Xincixiang’ pear has white and delicate flesh, juicy, sweet taste, unique aroma and excellent quality [41].’

  1. Comment: Figure1. It is almost impossible to distinguish between the symbols in the graph.

Response: Thanks for your kind suggestion. Figure1 has been adjusted.

  1. Comment: Line 181: use (P<0.05) instead of at the 0.05 level.

Response: Thanks for your kind suggestion. According to your suggestion, changes have been made at lines 162, 180 and 197 in the article.

  1. Comment: The Enlgish needs to be corected to a more scietific standard and checked for spelling mistakes (e.g. Line 16: I presume it should be bred instead of bed).

Response: Thanks for your kind suggestion. The bed in line 16 has been changed to bred, and the full text has been carefully proofread.

Best wishes,

Zhang Susu

Reviewer 2 Report

Overall, the article is well structured and appropriately written. There are some concerns.

In Introduction:

Line 59 “ACC” should be “ACS”

In Results

Figure 2: Figures 2b and c

Figure 3: Figures 3f, g, h and j

Figure 4 a-e

The figures should be shown as line graphs. Because, it presents information that needs to be seen as a trend of changes that occur over time.

Figure 4a should be an aldehyde, b should be an alcohol, and c should be an ester, according to lines 184-186.

Author Response

Response to Reviewer 2 Comments

Dear editor,

Thank you very much for your comments and suggestions. Those comments are all valuable and very helpful for revising and improving our paper, as well as the important guiding significance to our researches. We have studied comments carefully and have made correction which we hope meet with approval. The main corrections in the paper and the responds to the reviewer’s comments are as follows:

1.Comment: In Introduction: Line 59 “ACC” should be “ACS”.

Response: Thanks for your kind suggestion. ACC has been changed to ACS in the article.

2.Comment: In Results: Figure 2: Figures 2b and c; Figure 3: Figures 3f, g, h and j; Figure 4 a-e. The figures should be shown as line graphs. Because, it presents information that needs to be seen as a trend of changes that occur over time.

Response: Thanks for your kind suggestion. The figures listed have been changed to line graphs.

3.Comment: Figure 4a should be an aldehyde, b should be an alcohol, and c should be an ester, according to lines 184-186.

Response: Thanks for your kind suggestion. The figures in Figure 4 have been reordered.

Best wishes,

Zhang Susu